# Against the Resilience of High-Grade Gliomas: The Immunotherapeutic Approach (Part I)

**DOI:** 10.3390/brainsci11030386

**Published:** 2021-03-18

**Authors:** Alice Giotta Lucifero, Sabino Luzzi

**Affiliations:** 1Neurosurgery Unit, Department of Clinical-Surgical, Diagnostic and Pediatric Sciences, University of Pavia, 27100 Pavia, Italy; alicelucifero@gmail.com; 2Neurosurgery Unit, Department of Surgical Sciences, Fondazione IRCCS Policlinico San Matteo, 27100 Pavia, Italy

**Keywords:** bevacizumab, CAR T cell, cell-based therapy, glioblastoma, immunotherapy, malignant brain tumor, temozolomide

## Abstract

The resilience of high-grade gliomas (HGGs) against conventional chemotherapies is due to their heterogeneous genetic landscape, adaptive phenotypic changes, and immune escape mechanisms. Innovative immunotherapies have been developed to counteract the immunosuppressive capability of gliomas. Nevertheless, further research is needed to assess the efficacy of the immuno-based approach. The aim of this study is to review the newest immunotherapeutic approaches for glioma, focusing on the drug types, mechanisms of action, clinical pieces of evidence, and future challenges. A PRISMA (Preferred Reporting Items for Systematic Review and Meta-Analysis)-based literature search was performed on PubMed/Medline and ClinicalTrials.gov databases using the keywords “active/adoptive immunotherapy,” “monoclonal antibodies,” “vaccine,” and “engineered T cell.”, combined with “malignant brain tumor”, “high-grade glioma.” Only articles written in English published in the last 10 years were selected, filtered based on best relevance. Active immunotherapies include systemic temozolomide, monoclonal antibodies, and vaccines. In several preclinical and clinical trials, adoptive immunotherapies, including T, natural killer, and natural killer T engineered cells, have been shown to be potential treatment options for relapsing gliomas. Systemic temozolomide is considered the backbone for newly diagnosed HGGs. Bevacizumab and rindopepimut are promising second-line treatments. Adoptive immunotherapies have been proven for relapsing tumors, but further evidence is needed.

## 1. Introduction

High-grade gliomas (HGGs) are the most common malignant brain tumors with an incidence of 6/100,000 per year [1,2,3,4,5]. The current standard treatment protocol provides maximum surgical resection, adjuvant systemic chemotherapy, and whole-brain radiation [6,7]. The prognosis still remains extremely dismal, with a 5-year survival rate of less than 10% and a median survival of 14–16 months from diagnosis [8]. Intrinsic glioma cell heterogenicity, high mitotic activity, abnormal angiogenesis, and early local recurrence are responsible for the resilience of these tumors toward standard treatments [9,10,11]. Despite the biological complexity and adaptive nature of these lethal neoplasms, recent studies have identified some molecular and epigenetic markers, such as isocitrate dehydrogenase and O6-methylguanine-DNA-methyltransferase (MGMT) gene promoter, which are useful for predicting the prognosis and planning new targeted therapeutic options [12,13,14,15,16,17,18,19]. Moreover, the detection of glioma immunogenomics, alongside immunosuppressive mechanisms within the tumor microenvironment, has prompted the development of new immunotherapies [20,21,22,23,24,25,26,27].

The emerging immune-based technologies exploit manipulated molecules, purified tumor-specific neoantigens, and autologous/allogeneic lymphocytes, engineered for specific therapeutic purposes especially to counteract glioma-mediated immune suppression.

Therefore, the aim of the present study is to provide an overview of the classification, mechanisms of antitumor immune response, evidence from clinical trials, limitations, and future challenges of the immunotherapeutic approach for the treatment of malignant brain gliomas.

## 2. Materials and Methods

A comprehensive online literature review was performed in line with the Preferred Reporting Items for Systematic Reviews and Meta-Analysis (PRISMA) guidelines. The PubMed/Medline (https://pubmed.ncbi.nlm.nih.gov, accessed on 30 January 2021) and ClinicalTrials.gov (https://clinicaltrials.gov, accessed on 30 January 2021) databases were used, with combinations of Medical Subject Headings (MeSH) terms and text words. The main MeSH terms and key words were “malignant brain tumor,” “high-grade glioma,” and “glioblastoma”, further merged with “immunotherapy, active”; “immunotherapy, adoptive”. Supplementary research was conducted with additional MeSH terms: “chemotherapy”; “vaccine”; “alkylating agents”; “monoclonal antibodies”; “engineered T cell”; and “allogenic NK cell”, in order to restrict the field of interest to the novel immunotherapeutic strategies. The eligibility criteria included only articles written in English or translated, published in the last 10 years, and related to neuro-oncology. Review articles and editorials were included and filtered according to best match and relevance based on title and abstract.

On the ClinicalTrials.gov database, the search terms used to identify clinical trials were as follows: “malignant brain tumor,” “high-grade glioma” “central nervous system”, “immunotherapy, active”, and “immunotherapy, adoptive”. Interventional studies and clinical trials were included. No limits for the study phase or recruitment status were applied. Duplicates and titles with no English language translation available were removed. Trials related to innovative therapies for high-grade gliomas were chosen.

A descriptive analysis on the classification criteria, therapeutic mechanisms, and concluded and ongoing clinical trials was conducted. All the inclusion and exclusion criteria are outlined in Table 1.

## 3. Results

The literature search returned a total of 216 articles and 75 clinical trials. After the removal of duplicates and implementation of the exclusion criteria, a total of 135 articles and 69 clinical trials were assessed for eligibility and included in the review. Figure 1 presents the PRISMA flow chart for the literature selection process (Figure 1) and Table 2 summarizes the main clinical trials on immunotherapies for HGGs (Table 2).

### 3.1. Classification of Immunotherapies

Immunotherapies for brain gliomas can be classified based on the molecular mechanism and type of drug involved [27,28,29,30,31]. Two potential strategies were outlined: active immunotherapy, aimed at directly inducing the host antitumoral immune response, and adoptive (or passive) immunotherapy, resulting in the transduction of autologous/allogeneic immune cells aimed at the introduction or restoration of immune functions [29,32,33]. Alkylating agents, monoclonal antibodies, and vaccines are used for active immunotherapy, whereas engineered T cells, natural killer (NK) cells, and natural killer T (NKT) cells are used for adoptive immune strengthening. Table 3 reports the classification of novel immunotherapies for HGGs.

### 3.2. Active Immunotherapies

#### 3.2.1. Alkylating Agents

Temozolomide (TMZ) (Temodar^®^, Schering-Plough Research Institute, Kenilworth, NJ, USA), an oral alkylating agent, is the backbone of systemic therapy for HGGs [6,7]. Alkylating agents are basically chemotherapy drugs. They target tumor cells and block the cell cycle by damaging DNA double strands [34,35]. In prodrug status, TMZ crosses the blood–brain barrier (BBB) owing to its lipophilic properties and it reaches the tumor cells at therapeutic-relevant concentrations [36,37,38]. It is spontaneously converted into the active compound 5-(3-methyltriazen-1-yl)-imidazole-4-carboxamide (MTIC), which can methylate DNA bases. MTIC transfers alkyl groups to guanine or adenine at the N7-/O6- or N3-position, respectively. The methylation results in base pair mismatch and chromatin structure remodeling, induces G2/M cell cycle arrest, and finally leads to cell apoptosis [37,39,40,41]. As a result, TMZ therapeutic success is closely dependent on DNA repair pathways, which leads to glioma therapeutic resistance [42,43]. The demethylating enzyme MGMT acts directly in repairing O6-methylguanine, consequently avoiding gene mutation and glioma cell death [44,45]. The methylation status of the MGMT promoter and the enzyme expression predict the response to TMZ and are also prognostic factors of survival [46,47,48,49]. Figure 2 describes the TMZ mechanism of action and its effects in the glioma cells (Figure 2).

#### 3.2.2. Monoclonal Antibodies

Monoclonal antibodies (MAbs) are purified immunoglobulin with monovalent affinity. MAbs bind specific molecular epitopes and enlist the host immune system against tumors. Bevacizumab (BVZ) is the most widely tested, especially for the treatment of recurrent glioblastoma [50,51,52,53,54,55,56,57]. It directly targets the vascular endothelial growth factor A (VEGF-A) inhibiting the interaction with VEGF tyrosine kinase receptor, which is overexpressed on the surface of endothelial cells as a result of tissue hypoxia, and interrupts aberrant tumor angiogenesis (Figure 3A). The antiangiogenic effect exerted by BVZ extends to the reduction of microvessel density and vascular permeability, resulting in the antiedema effect [51,58,59]. Furthermore, recent studies have investigated the potential role of BVZ in modulating the immunosuppressive tumor microenvironment [60,61,62]. Two phase III clinical trials, the Avastin in Glioblastoma (AVAglio92) [63] (#NCT00943826) and the Radiation Therapy Oncology Group (RTOG)-082593 [64] (#NCT00884741), studied the combination of BVZ with standard therapy, demonstrating promising results for recurrent HGGs.

The AVAglio92 documented a significant improvement in the median progression free survival (PFS) to 10.6 months in the BVZ group, versus 6.2 months in the placebo one (*p* < 0.001). Overall survival (OS) proved to be higher only during the first two years of treatment in the BVZ group (72.4% versus 66.3%) [63]. The RTOG-082593 reported an increase in PFS in the BVZ group compared to the control group (10.7 months versus 7.3 months, *p* = 0.007), but equally failed to demonstrate a better OS [64].

Nivolumab, a human IgG4, directly binds programmed cell death protein 1 (PD-1), exposed on activated T cells. Physiologically, PD-1 interacts with its ligands, namely PD-L1 and PD-L2, downregulating the immune cascade. PD-Ls are overexpressed on HGG cells as a mechanism of immune escape. Nivolumab impounds PD-1 and boosts the antitumoral activity of CD4+ and CD8+ cells (Figure 3B) [65,66,67]. Despite BVZ and nivolumab being studied for glioma therapy in several clinical trials and also as a combined protocol (#NCT03890952, #NCT03743662, #NCT03452579) [68,69,70], the administration route remains still a concern. Because the BBB physiologically blocks the antibody access to the brain, several studies are focusing on innovative mAb-delivering routes, to improve the drug efficacy and intratumoral uptake. Intra-arterial administration, intracranial injection, and nanoparticle and liposomal carriers are currently potential strategies [56,71,72,73,74].

#### 3.2.3. Vaccine

Anticancer vaccination was designed to strengthen host immune response by stimulating the production of self-antibodies against tumor antigens. Advanced research in translational medicine has allowed the isolation of specific glioma immunogens, i.e., epidermal growth factor receptor variant III (EGFRvIII). EGFRvIII is an active splice variant, found in 30% of primary glioblastoma (GBM), which enhances cell proliferation and survival [75,76,77,78,79,80]. Rindopepimut (Rintega^®^, Celldex Therapeutics, Inc., Phillipsburg, NJ, USA), a 14-mer injectable peptide vaccine against EGFRvIII, was projected to activate CD4+ and CD8+ T cells against malignant brain tumor cells (Figure 4). It was also evaluated in some preclinical studies. The first phase I trial, VICTORI, showed an excellent safety profile, a PFS of 10.2 months and an OS of 22.8 months. Some phase II trials (ACTIVATE, ACT II–III, and ReACT) confirmed the low toxicity and demonstrated an increase in median progression-free and overall survival for recurrent gliomas. The ACTIVATE phase II trial reported a PFS of 14.2 and 6.4 months, and an OS of 26 and 15.2 months in the vaccinated patients and control group, respectively. In ACT II and III studies, the PFS was 15.2 and 12.3 for the rindopepimut group, respectively, while the PFS was 6.4 months in both control groups. The OS was 23.6 and 24.6 months in ACT II and III trials, respectively, compared to the control groups (15.2 months). ReACT showed a PFS at 6 months of 28% for rindopepimut, compared to 16% in the control group [79,81,82,83,84].

ACT IV, a multicentric phase III trial, tested the combination of rindopepimut and standard chemotherapy with TMZ in newly diagnosed GBM, showing no significant difference in OS between the two groups (#NCT01480479) [85].

Several other multiple-epitope vaccines are also under development for glioma treatment [86]. Among these vaccines, the dendritic cell vaccine (DCVax-Brain) consists of purified dendritic cells and tumor antigens (#NCT00045968) [87,88], whereas the personalized cellular vaccine (PerCellVac2) is made up of allogeneic peripheral blood cells combined with autologous glioma antigens (#NCT02808364, #NCT02709616) [89].

### 3.3. Adoptive Immunotherapies

#### 3.3.1. Engineered T Cells

In the field of adoptive immunotherapies, T-cell-based strategies are the most promising. Protocols begin with patient leukapheresis, followed by the engineering of autologous T cells via viral vectors. The viral transduction integrates autologous T cells with tumor-specific antigen receptor genes, specifically the chimeric antigen receptor (CAR) and the transgenic T cell receptor (TCR) genes [33,90]. CARs have an extracellular domain that binds tumor-specific ligands, whereas the intracellular domain activates T cell pathways [91,92,93,94,95,96,97]. The specificity of CAR T depends on the transinfected CAR genes. Several CAR genes were investigated: EGFRvIII (#NCT02209376, #NCT02664363, #NCT03726515), human epidermal growth factor receptor 2 (HER2) (#NCT01109095, #NCT03389230), interleukin-13 receptor A2 (IL13Ra2) (#NCT00730613, #NCT02208362, #NCT04003649), and erythropoietin-producing hepatocellular carcinoma A2 (EphA2). EGFRvIII-, HER2-, and IL13Ra2-targeted CAR T cells were just tested for HGG therapy in preclinical and clinical trials. In 2017, O’Rourke and colleagues treated 10 patients with a single-dose of EGFRvIII CAR T cells (#NCT02209376), reporting no dose-related side effects and a good safety profile, but neither the OS nor the PFS was shown to increase. In 2017, Ahmed et al. tested HER2-specific CAR T cells for treatment of 17 recurrent HER2+ high-grade gliomas (#NCT01109095) and showed similar results. Further studies are needed to determine their feasibility and safety [98,99,100,101,102,103,104,105,106,107].

The TCR complex is exposed on the surface of T cells and interacts with the major histocompatibility complex (MHC), leading to activation of the cellular immune cascade. TCR transgenic T cells are genetically engineered to express receptors against specific tumor MHC and stimulate the immune response against malignant brain tumors [108,109,110,111]. CAR and TCR engineered T cells are expanded ex vivo and injected, cross the BBB, bind the tumor cells, and activate the antitumoral immune cascade, thus promoting apoptosis (Figure 5) [94,112,113,114,115].

#### 3.3.2. NK Cells

NK cells are involved in the treatment of malignant brain tumors by means of different strategies [116]. The first strand can exploit allogeneic NK cells and target tumor cells by the lack of MHC and, without being inactivated, exert oncolytic effects [117,118,119,120]. Another approach includes the NK immunoglobulin-like receptor antibodies (anti-KIR Abs). They block the link between NK cells and MHC class I, avoid the immunosuppression response, and increase the NK-mediated tumor apoptosis [116,121]. Furthermore, the antibody-dependent cell-mediated cytotoxicity (ADCC) is a mechanism wherein specific antibodies, directed to HGG antigens, interact with NK cells, promoting tumor lyses. The Fab portion binds antigens on the tumor cell surface, whereas the fragment crystallizable (Fc) region recognizes the NK cell receptors, such as CD16/FcγRIII, mediating the antitumoral immune response. Among ADCC, EGFR tumor antigen and CD16 (FcγIIIA), KIR2DS2, and NK Group 2D (NKG2D) receptors are the most researched [121].

In 2017, Yvon and colleagues proved the efficacy of cord blood NK cells that were retrovirally transduced and manipulated to express the TGF-β-dominant-negative receptor II (DNRII) [121,122]. DNRII makes NK cells immune to the TGF-β that is overexpressed in the tumor microenvironment as an immune evasion promoter. Other techniques under development involve the exosomes as a vector for NK cells and a new sort of CAR (CAR-KHYG-1) targeting EGFRvIII [123,124,125].

#### 3.3.3. NKT Cells and Hybrid Therapies

NKT cells are a subgroup of T cells with concomitant properties of T and NK cells, both expressing the type of molecular surfaces. Particularly, CD1d-restricted NKT cells were found to play an important role as immunoregulatory cells within the glioma microenvironment [126]. Strategies to overcome the immune tolerance of glioma include the expansion in vitro of autologous mature dendritic cells (DCs) with NKT ligand α-galactosyl ceramide, which enhances NKT cell cytotoxic activity [127,128]. Another integrated approach is the autologous lymphoid effector cells specific against tumor cell (ALECSAT) therapy (Cytovac A/S, Hørsholm, Denmark). ALECSAT is a 26-day immunization protocol. Autologous lymphocytes and monocytes were isolated and differentiated in vitro. The mature DCs and nonactivated lymphocytes are cultured with CD4+ T cells and 5-aza-2′-deoxycytidine, a DNA-demethylation agent, which induces the expression of tumor antigens. The activated CD4+ T cell and CD8+ cytotoxic lymphocyte selectively target tumor cells, whereas the glioma cells that do not express the antigen are targeted by activated NK cells, reducing the chance of cancer immune evasion (#NCT02060955) [129].

## 4. Discussion

The present study reviews current immunotherapeutic approaches to malignant brain tumors, specifically focusing on classification, immunomechanisms, evidence from clinical trials, limitations, and future challenges.

The current first-line multimodal treatment involves the combination of gross total surgical resection, followed by the use of adjuvant alkylating agents and whole-brain radiotherapy [6,7,130]. Despite TMZ being the only systemic agent approved as of now, glioma cells have often been found to be resistant [131,132,133]. One of the major disadvantages of TMZ is the rapid in vivo hydrolytic degradation. Several studies seek to overcome the quick drug clearance through polymer scaffold molecules, simultaneously upgrading therapeutic efficacy and reducing adverse events [134,135]. TMZ also frequently induces lymphopenia and myelosuppression, as do many other chemotherapy drugs. The immunodepletion leads to less immune surveillance and inhibition of antitumor immune response [136,137,138]. Another downside is the tumor expression of DNA repair enzymes, such as MGMT, which nullify the alkylating agents’ therapeutic effects. The MGMT promoter status heavily influences overall survival, which is almost 21 months for methylated MGMT HGGs rather than 13 months for the unmethylated ones [46,49,139,140].

The failure of current therapeutic approaches, combined with advances in translational medicine, has led to the development of new strategies. The first turning point was the use of MAbs and BVZ. The Food and Drug Administration approved the use of BVZ based on the encouraging results achieved by the phase II BRAIN study (#NCT00345163), conducted on 167 patients affected by recurrent GBMs [141]. The initial enthusiasm around BVZ approval waned due to the lack of an effective overall survival improvement. The decrease of enhancement on MRI T1, which occurred in up to 60% of BVZ patients and was defined as a radiographic pseudo-response, is nothing more than an antiangiogenic effect without real tumor shrinkage. BVZ treatment stops blood vessel proliferation and reduces BBB permeability, resulting in an immediate contrast-enhancement decrease. However, these microvascular changes do not affect tumor proliferation or overall survival [142,143,144,145]. The third therapeutic option, within active immunotherapies, is the antitumoral vaccination. Peptide vaccines are designed to activate host immunity versus tumor-specific antigens. They have proven to be feasible and safe, but are still ineffective [80,146]. HGGs’ intrinsic heterogeneity, mechanisms of immune escape, and loss of antigenic variants mediate patient immunosuppression and invalidate the potency of the vaccine. Personalized multipeptide vaccines that are customized to target more than one tumor antigen should be developed in the future [147,148].

The glioma microenvironment, populated by cancer stem cells, immunosuppressive molecules, and interleukins, plays a pivotal role in tumor growth and immune resistance [20,149,150,151,152,153,154]. The rationale of adoptive immunotherapies is to modulate the tumor setting through the activation of cell immunity. These technologies are based on the manipulation of autologous/allogenic immunological cells and the engineering of CD8+ and CD4+ T lymphocytes [155]. Preliminary studies have considered CAR T cells as a valid and safe tool, which are tested intravenously and through intracranial infusion. However, the limited survival of T CAR, mechanisms of antigen escape, and ineffective homing to tumor cells do not make this a strategy yet achievable as a second-line treatment. The emerging NK-cell-based approach still has its limitations, such as the high expression of MHC I and HLA type A on glioma cells, which tie the KIRs and impede NK cells’ antitumoral activity.

### Limitations and Future Perspectives

Several aspects are responsible for the resilience of glioma, such as the lack of tumor antigen, loss of immunological phenotype, and immune evasive molecule production, which all together still limit the success of both active and adoptive immunotherapies.

Furthermore, the main limitation lies in the need to overcome the BBB, the reason why the administration route is pivotal. Several studies have proposed innovative methods, such as the implanted intracerebral convection-enhanced deliveries or the superselective intra-arterial cerebral infusion [156,157,158,159,160,161]. The intra-arterial administration is additionally facilitated by the previous destruction of BBB, carried out through osmotic agents, hypertonic solutions and mannitol, or ultrasounds [156,162,163]. Low frequency ultrasounds are a non-invasive strategy which lead to BBB disruption resulting in increased drug penetration and faster time to reach the tumor site [162]. An innovative emerging technique exploits the vascularized temporoparietal fascial flaps with the aim to bypass the BBB. These flaps are vascularized by the external carotid artery system, free from the BBB system. They can be transposed into the surgical cavity, after glioma resection, and could allow effective drug penetration and residual tumor cell targeting [164].

Another key aspect is the engineering of biocompatible and non-toxic small carriers able to improve drug diffusion and tissue distribution. Viral vehicles, nanoparticles, and liposomes are currently the most investigated [165,166,167,168].

The development of new administration routes and the advances in engineering more efficient and safe carriers will allow the implementation of standard therapeutic protocols with concomitant tailored immunotherapies.

## 5. Conclusions

Immunotherapeutic strategies are designed to overcome the immune escape pathways and the immunosuppressive glioma microenvironment, which are responsible for the resilience of HGGs.

Antitumoral immunotherapies involve active immune products along with adoptive engineered T, NK, and NKT cells.

BVZ is indicated in the treatment of recurrent HGGs, but not yet within the upfront protocol for newly diagnosed HGGs. Vaccines have been proven to be safe and feasible, although their treatment efficacy needs to be evaluated further. CAR T cells against EGFRvIII and engineered TCR T, NK, and hybrid cells have demonstrated a promising potential in several preclinical studies.

Although they are still not considered to be the first-line treatment against malignant gliomas, immunotherapies have shown excellent results in improving PFS.

Further clinical studies are focusing on validating antitumoral immunotherapeutic approaches such as personalized second-line strategies.

## Figures and Tables

**Figure 1 brainsci-11-00386-f001:**
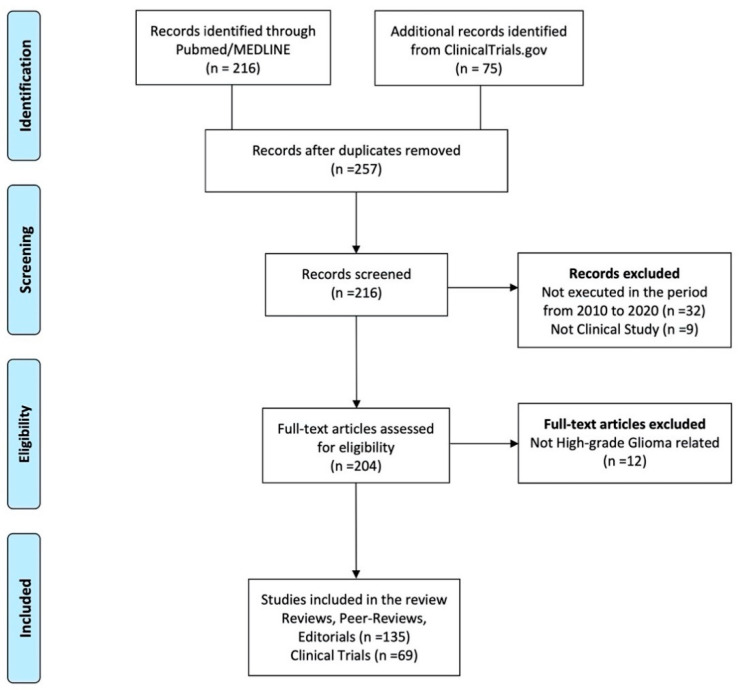
PRISMA (Preferred Reporting Items for Systematic Reviews and Meta-Analysis) flow-chart.

**Figure 2 brainsci-11-00386-f002:**
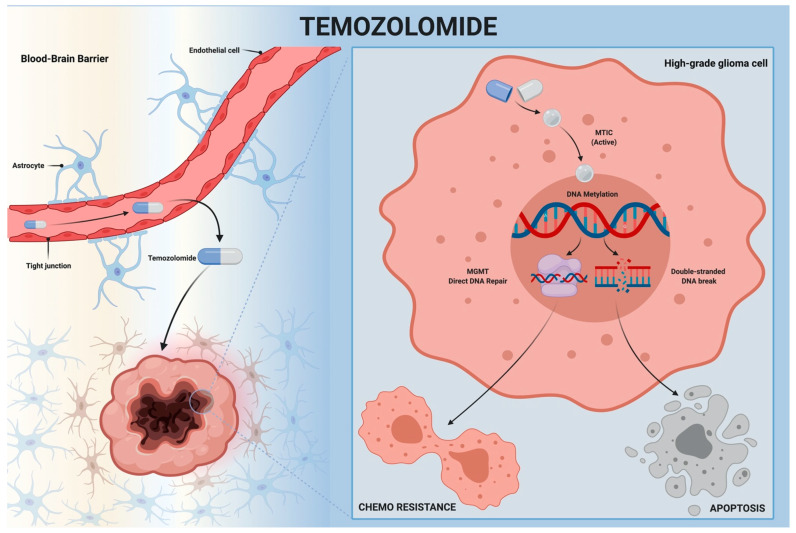
Temozolomide mechanism of action. Temozolomide crosses the blood–brain barrier and reaches high-grade glioma cells. It is spontaneously converted to the active compound (MTIC), which methylates DNA bases, breaks double-strand DNA, and leads to apoptosis. MTIC action is opposed to the DNA repair pathways, including MGMT. MGMT: O6-methylguanine-DNA methyltransferase; MTIC: 5-(3-methyltriazen-1-yl)-imidazole-4-carboxamide; TMZ: Temozolomide.

**Figure 3 brainsci-11-00386-f003:**
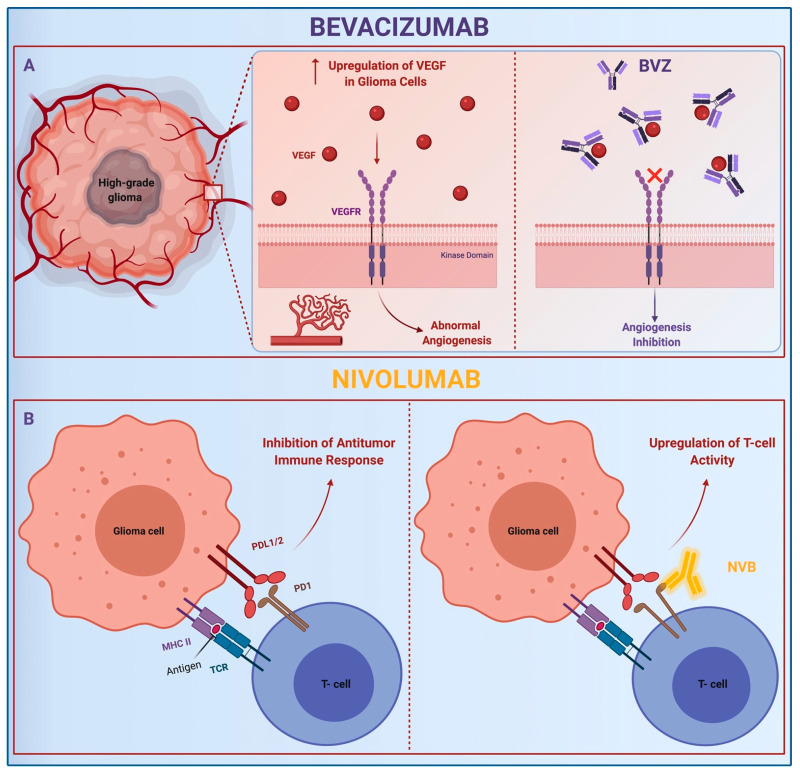
Mechanism of action of MAbs, bevacizumab and nivolumab. BVZ: (**A**) Bevacizumab; MHC: Major Histocompatibility Complex; NVB: (**B**) Nivolumab; PD-1: Programmed Cell Death Protein. Abbreviations: PDL-1/2: Programmed Cell Death Protein Ligand 1/2; TCR: Transgenic T Cell Receptor; VEGF-A: Vascular Endothelial Growth Factor A; VEGFR: Vascular Endothelial Growth Factor Tyrosine Kinases Receptor.

**Figure 4 brainsci-11-00386-f004:**
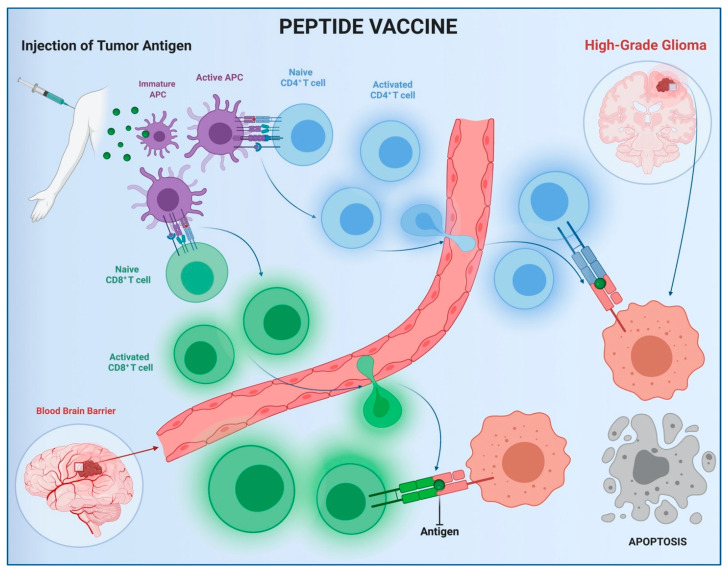
Peptide vaccine mechanism of action. APC: Antigen-Presenting Cell.

**Figure 5 brainsci-11-00386-f005:**
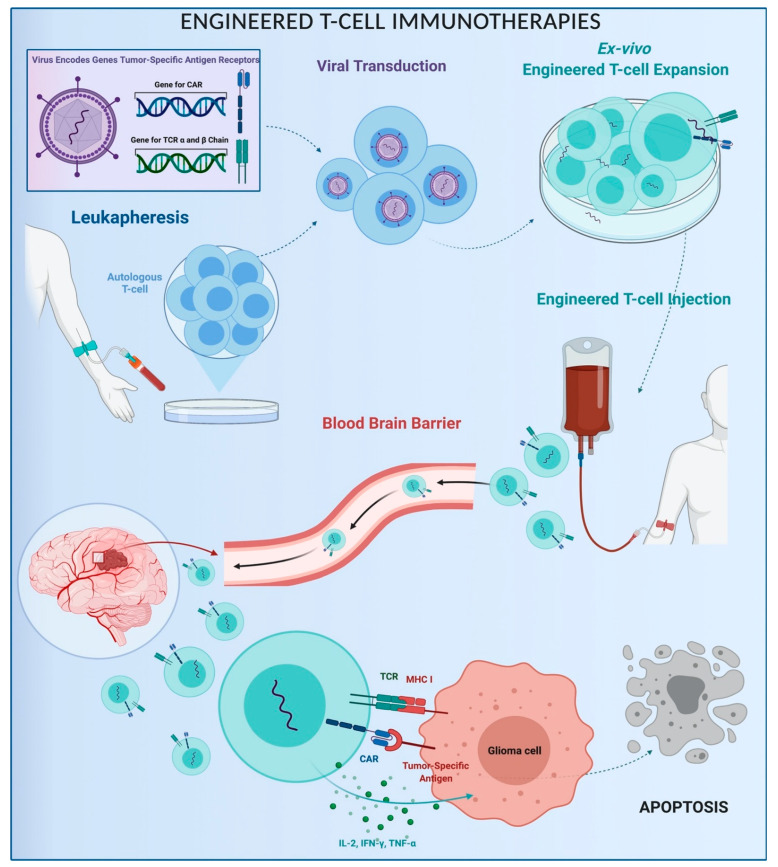
Schematic representation of engineered T-cell immunotherapy. CAR: Chimeric Antigen Receptor; MHC: Major Histocompatibility Complex; TCR: Transgenic T Cell Receptor.

**Table 1 brainsci-11-00386-t001:** Inclusion and exclusion criteria for systematic review.

Inclusion Criteria	Exclusion Criteria
Reviews, peer-reviews, editorials	Case reports, abstracts, and dissertations
Clinical, pre-clinical Trials	Withdrawn or abandoned clinical trials
English language, or translated	Non-English language
Publications from 2010–2020	Studies not from 2010–2020
Studies on humans or human products	Animal studies
Publications related to neuro-oncology	Publications not related to neuro-oncology
Publications related to high-grade glioma	Publications not related to high-grade glioma
Adult and pediatric patients	

**Table 2 brainsci-11-00386-t002:** Main clinical trials on immunotherapies for high-grade gliomas.

#	ClinicalTrials.gov Identifier	Title	Status	Phase	Diseases	# of Pts. Enrolled	Treatment	Locations
1	NCT03011671	Study of Acetazolamide With Temozolomide in Adults With Newly Diagnosed or Recurrent Malignant Glioma	Suspended	I	Malignant Glioma of Brain	24	Acetazolamide, Temozolomide	USA
2	NCT02416999	Ultra-low Dose Bevacizumab Plus Temozolomide for Recurrent High-grade Gliomas	Unknown	NA	Recurrent High-grade Glioma	30	Ultra-low dose Bevacizumab, Temozolomide	CHN
3	NCT01891747	A Phase I Study of High-dose L-methylfolate in Combination With Temozolomide and Bevacizumab in Recurrent High Grade Glioma	Active, not recruiting	I	Malignant Glioma	12	Bevacizumab, Temozolomide, Vitamin C	USA
4	NCT04267146	Nivolumab in Combination With Temozolomide and Radiotherapy in Children and Adolescents With Newly Diagnosed High-grade Glioma	Recruiting	I, II	High Grade Glioma	40	Nivolumab, Temozolomide, Radiotherapy	FR
5	NCT00782756	Bevacizumab, Temozolomide and Hypofractionated Radiotherapy for Patients With Newly Diagnosed Malignant Glioma	Completed	II	Brain Cancer, Malignant Glioma	40	Radiotherapy, Temozolomide, Bevacizumab	USA
6	NCT04547621	HSRT and IMRT Chemoradiotherapy for Newly Diagnosed GBM	Active, not recruiting	I, II	Glioma, Malignant	50	Radiation, Temozolomide	CHN
7	NCT01390948	A Study of Bevacizumab (Avastin) in Combination With Temozolomide and Radiotherapy in Paediatric and Adolescent Participants With High-Grade Glioma	Completed	II	High Grade Glioma	124	Bevacizumab, Radiotherapy, Temozolomide	A
8	NCT00660621	A Phase II Study Of Gliadel, Concomitant Temozolomide And Radiation, Followed By Dose Dense Therapy With Temozolomide Plus Bevacizumab For Newly Diagnosed Malignant High Grade Glioma	Unknown	II	Glioma	40	Temozolomide, Bevacizumab	USA
9	NCT03633552	Efficacy of Two Temozolomide Regimens in Adjuvant Treatment of Patients With Brain High Grade Glioma	Recruiting	III	Glioblastoma Multiforme Anaplastic Astrocytoma	62	Temozolomide	IR
10	NCT01105702	Temodar (Temozolomide), Bevacizumab, Lithium and Radiation for High Grade Glioma	Terminated	II	Brain Cancer	28	Temozolomide, Bevacizumab, Lithium Carbonate, Radiation	USA
11	NCT01740258	Bevacizumab Beyond Progression (BBP)	Completed	II	Malignant Glioma, Glioblastoma, Gliosarcoma	68	Radiation Therapy, Temozolomide, Bevacizumab	USA
12	NCT01478321	Efficacy of Hypofractionated XRT w/Bev. + Temozolomide for Recurrent Gliomas	Terminated	II	Adult Anaplastic Astrocytoma Ependymoma Oligodendroglioma/Glioblastoma	54	Temozolomide, Bevacizumab Hypofractionated radiation therapy	USA
13	NCT00943826	A Study of Bevacizumab (Avastin^®^) in Combination With Temozolomide and Radiotherapy in Participants With Newly Diagnosed Glioblastoma	Completed	III	Glioblastoma	921	Bevacizumab, Temozolomide Radiation therapy	USA
14	NCT00884741	Temozolomide and Radiation Therapy With or Without Bevacizumab in Treating Patients With Newly Diagnosed Glioblastoma	Completed	III	Glioblastoma, Gliosarcoma, Supratentorial Glioblastoma	637	Radiation Therapy, Temozolomide, Bevacizumab	USA
15	NCT01046279	Hypertension Monitoring in Glioma Patients Treated With Bevacizumab	Terminated	NA	Glioma	40	Bevacizumab	ZH
16	NCT00271609	Bevacizumab for Recurrent Malignant Glioma	Completed	II	Recurrent High-Grade Gliomas	88	Bevacizumab	USA
17	NCT02833701	Bevacizumab and Ascorbic Acid in Patients Treating With Recurrent High Grade Glioma	Terminated	I	Glioblastoma, Glioma	9	Ascorbic Acid, Bevacizumab	USA
18	NCT00595322	Bevacizumab in the Radiation Treatment of Recurrent Malignant Glioma	Completed	NA	Recurrent Malignant Gliomas, Primary Brain Tumor	25	Bevacizumab, Radiation	USA
19	NCT00337207	Bevacizumab in Treating Patients With Recurrent or Progressive Glioma	Completed	II	Central Nervous System Tumors	55	Bevacizumab	USA
20	NCT01091792	Exploratory Study of the Modulation of the Immune System by VEGF Blockade in Patients With Glioblastoma Multiforme (GBM)	Completed	I	Glioblastoma Multiforme	13	Bevacizumab	USA
21	NCT00883298	Bi-weekly Temozolomide Plus Bevacizumab for Adult Patients With Recurrent Glioblastoma Multiforme	Completed	II	Recurrent Glioblastoma Multiforme Recurrent Gliosarcoma	30	Temozolomide, Bevacizumab	USA
22	NCT01811498	Repeated Super-Selective Intraarterial Cerebral Infusion of Bevacizumab (Avastin) for Treatment of Newly Diagnosed GBM	Active, not recruiting	I, II	Glioblastoma Multiforme, Brain Tumor	25	Bevacizumab	USA
23	NCT01730950	Bevacizumab With or Without Radiation Therapy in Treating Patients With Recurrent Glioblastoma	Active, not recruiting	II	Adult Giant Cell Glioblastoma, Glioblastoma, Adult Gliosarcoma Recurrent Adult Brain Tumor	182	Bevacizumab, Radiation therapy	USA
24	NCT02761070	Bevacizumab Alone Versus Dose-dense Temozolomide Followed by Bevacizumab for Recurrent Glioblastoma, Phase III	Recruiting	III	Recurrent Glioblastoma	146	Temozolomide, Bevacizumab	J
25	NCT01209442	Hypofractionated Intensity-Modulated Radiation Therapy With Temozolomide and Bevacizumab for Glioblastoma Multiforme	Completed	II	Glioblastoma Multiforme	30	Bevacizumab, Temozolomide Radiation Therapy	USA
26	NCT01125046	Bevacizumab in Treating Patients With Recurrent or Progressive Meningiomas	Completed	II	Central Nervous System Tumors	50	Bevacizumab	USA
27	NCT01526837	Bevacizumab (Avastin) Into the Tumor Resection Cavity in Subjects With Glioblastoma Multiforme at First Recurrence	Terminated	I	Glioblastoma Multiforme	1	Bevacizumab	USA
28	NCT01443676	Avastin Plus Radiotherapy in Elderly Patients With Glioblastoma	Completed	II	Glioblastoma	75	Bevacizumab, Radiation therapy	USA
29	NCT00590681	Bevacizumab and Temozolomide Following Radiation and Chemotherapy for Newly Diagnosed Glioblastoma Multiforme	Completed	II	Glioblastoma Multiforme	62	Bevacizumab, Temozolomide	USA
30	NCT03925246	Efficacy of Nivolumab for Recurrent IDH Mutated High-Grade Gliomas	Active, not recruiting	II	High Grade Glioma, Brain Cancer	43	Nivolumab	FR
31	NCT00345163	A Study to Evaluate Bevacizumab Alone or in Combination With Irinotecan for Treatment of Glioblastoma Multiforme (BRAIN)	Completed	II	Glioblastoma	167	Bevacizumab, Irinotecan	NA
32	NCT03890952	Translational Study of Nivolumab in Combination With Bevacizumab for Recurrent Glioblastoma	Recruiting	II	Recurrent Adult Brain Tumor	40	Nivolumab, Bevacizumab	DNK
33	NCT01498328	A Study of Rindopepimut/GM-CSF in Patients With Relapsed EGFRvIII-Positive Glioblastoma	Completed	II	Glioblastoma	127	Rindopepimut (CDX-110) with GM-CSF Bevacizumab, KLH	USA
34	NCT03743662	Nivolumab With Radiation Therapy and Bevacizumab for Recurrent MGMT Methylated Glioblastoma	Recruiting	II	Glioblastoma	94	Re-irradiation, Bevacizumab Nivolumab, Re-resection	USA
35	NCT03452579	Nivolumab Plus Standard Dose Bevacizumab Versus Nivolumab Plus Low Dose Bevacizumab in GBM	Active, not recruiting	II	Glioblastoma	90	Nivolumab, Standard/Reduced Dose Bevacizumab	USA
36	NCT02550249	Neoadjuvant Nivolumab in Glioblastoma	Completed	II	Glioblastoma Multiforme	29	Nivolumab	ES
37	NCT04195139	Nivolumab and Temozolomide Versus Temozolomide Alone in Newly Diagnosed Elderly Patients With GBM	Recruiting	II	Glioblastoma Multiforme	102	Nivolumab, Temozolomide	A
38	NCT02667587	An Investigational Immuno-therapy Study of Temozolomide Plus Radiation Therapy With Nivolumab or Placebo, for Newly Diagnosed Patients With Glioblastoma (GBM, a Malignant Brain Cancer)	Active, not recruiting	III	Brain Neoplasms	693	Nivolumab, Temozolomide, Radiotherapy	USA
39	NCT02617589	An Investigational Immuno-therapy Study of Nivolumab Compared to Temozolomide, Each Given With Radiation Therapy, for Newly-diagnosed Patients With Glioblastoma (GBM, a Malignant Brain Cancer)	Active, not recruiting	III	Brain Cancer	560	Nivolumab, Temozolomide, Radiotherapy	USA
40	NCT01213407	Dendritic Cell Cancer Vaccine for High-grade Glioma	Completed	II	Glioblastoma Multiforme	87	Trivax, Temozolomide, Surgery, Radiotherapy	AU
41	NCT02529072	Nivolumab With DC Vaccines for Recurrent Brain Tumors	Completed	I	Malignant Glioma, Astrocytoma	6	Nivolumab	USA
42	NCT03718767	Nivolumab in Patients With IDH-Mutant Gliomas With and Without Hypermutator Phenotype	Recruiting	II	Malignant Glioma of Brain	95	Nivolumab	USA
43	NCT01480479	Phase III Study of Rindopepimut/GM-CSF in Patients With Newly Diagnosed Glioblastoma	Completed	III	Malignant Glioma of Brain	745	Rindopepimut (CDX-110) with GM-CSF Temozolomide, KLH	USA
44	NCT01058850	Phase I Rindopepimut After Conventional Radiation in Children w/Diffuse Intrinsic Pontine Gliomas	Terminated	I	Brain Cancer, Brain Stem Tumors	3	Rindopepimut	USA
45	NCT00045968	Study of a Drug [DCVax^®^-L] to Treat Newly Diagnosed GBM Brain Cancer	Unknown	III	Glioblastoma	348	Dendritic cell immunotherapy	USA
46	NCT02808364	Personalized Cellular Vaccine for Recurrent Glioblastoma (PERCELLVAC2)	Active, not recruiting	I	Glioblastoma	10	Personalized cellular vaccine	CHN
47	NCT02709616	Personalized Cellular Vaccine for Glioblastoma (PERCELLVAC)	Active, not recruiting	10	Glioblastoma	10	Personalized cellular vaccine	CHN
48	NCT02209376	Autologous T Cells Redirected to EGFRVIII-With a Chimeric Antigen Receptor in Patients With EGFRVIII+ Glioblastoma	Terminated	I	Patients With Residual or Reccurent EGFRvIII+ Glioma	11	CART-EGFRvIII T cells	USA
49	NCT03726515	CART-EGFRvIII + Pembrolizumab in GBM	Active, not recruiting	I	Glioblastoma	7	CART-EGFRvIII T cells, Pembrolizumab	USA
50	NCT02664363	EGFRvIII CAR T Cells for Newly-Diagnosed WHO Grade IV Malignant Glioma	Terminated	I	Glioblastoma, Gliosarcoma	3	EGFRvIII CAR T cells	USA
51	NCT01109095	CMV-specific Cytotoxic T Lymphocytes Expressing CAR Targeting HER2 in Patients With GBM	Completed	I	Glioblastoma Multiforme	16	HER.CAR CMV-specific CTLs	USA
52	NCT02208362	Genetically Modified T-cells in Treating Patients With Recurrent or Refractory Malignant Glioma	Recruiting	I	Glioblastoma, Recurrent Malignant Glioma	92	IL13Ralpha2-specific Hinge-optimized 41BB-co-stimulatory CAR Truncated CD19-expressing Autologous T-Lymphocytes	USA
53	NCT04003649	IL13Ralpha2-Targeted Chimeric Antigen Receptor (CAR) T Cells With or Without Nivolumab and Ipilimumab in Treating Patients With Recurrent or Refractory Glioblastoma	Recruiting	I	Recurrent/Refractory Glioblastoma	60	IL13Ralpha2-specific Hinge-optimized 4-1BB-co-stimulatory CAR/Truncated CD19-expressing Autologous TN/MEM Cells, Ipilimumab, Nivolumab	USA
54	NCT03383978	Intracranial Injection of NK-92/5.28.z Cells in Patients With Recurrent HER2-positive Glioblastoma	Recruiting	I	Glioblastoma	30	NK-92/5.28.z	DE
55	NCT03392545	Combination of Immunization and Radiotherapy for Malignant Gliomas (InSituVac1)	Recruiting	I	High Grade Glioma, Glioblastoma, Glioma of Brainstem	30	Combined immune adjuvants and radiation	CHN
56	NCT03389230	Memory-Enriched T Cells in Treating Patients With Recurrent or Refractory Grade III-IV Glioma	Recruiting	I	High Grade Glioma, Glioblastoma	42	HER2(EQ)BBζ/CD19t+ Tcm	USA
57	NCT03347097	Adoptive Cell Therapy of Autologous TIL and PD1-TIL Cells for Patients With Glioblastoma Multiforme	Active, not recruiting	I	Glioblastoma Multiforme	40	TIL	CHN
58	NCT03344250	Phase I EGFR BATs in Newly Diagnosed Glioblastoma	Recruiting	I	Glioblastoma Multiforme	18	EGFR BATs with SOC RT and TMZ	USA
59	NCT03170141	Immunogene-modified T (IgT) Cells Against Glioblastoma Multiforme	Enrolling by invitation	I	Glioblastoma Multiforme	20	Antigen-specific IgT cells	CHN
60	NCT02937844	Pilot Study of Autologous Chimeric Switch Receptor Modified T Cells in Recurrent Glioblastoma Multiforme	Unknown	I	Glioblastoma Multiforme	20	Anti-PD-L1 CSR T cells, Cyclophosphamide, Fludarabine	CHN
61	NCT02799238	Autologuos Lymphoid Effector Cells Specific Against Tumour (ALECSAT) as Add on to Standard of Care in Patients With Glioblastoma	Completed	II	Glioblastoma	62	ALECSAT, Radiotherapy, Temozolomide	SE
62	NCT02060955	Randomized Phase 2 Study to Investigate Efficacy of ALECSAT in Patients With GBM Measured Compared to Avastin/Irinotecan	Terminated	II	Glioblastoma Multiforme	25	ALECSAT, Bevacizumab/Irinotecan	DK
63	NCT01588769	A Phase I Study to Investigate Tolerability and Efficacy of ALECSAT Administered to Glioblastoma Multiforme Patients	Completed	I	Glioblastoma Multiforme	23	ALECSAT cell based immunotherapy	DK
64	NCT01454596	CAR T Cell Receptor Immunotherapy Targeting EGFRvIII for Patients With Malignant Gliomas Expressing EGFRvIII	Completed	I, II	Malignant Glioma, Glioblastoma, Gliosarcoma	18	Epidermal growth factor receptor(EGFRv)III Chimeric antigen receptor (CAR) transduced PBL, Aldesleukin, Fludarabine, Cyclophosphamide	USA
65	NCT01144247	Cellular Immunotherapy Study for Brain Cancer	Completed	I	Malignant Glioma of Brain	10	Alloreactive CTL	USA
66	NCT01082926	Phase I Study of Cellular Immunotherapy for Recurrent/Refractory Malignant Glioma Using Intratumoral Infusions of GRm13Z40-2, An Allogeneic CD8+ Cytolytic T-Cell Line Genetically Modified to Express the IL 13-Zetakine and HyTK and to be Resistant to Glucocorticoids, in Combination With Interleukin-2	Completed	I	Malignant Glioma of Brain	6	Therapeutic allogeneic lymphocytes, Aldesleukin	USA
67	NCT00730613	Cellular Adoptive Immunotherapy Using Genetically Modified T-Lymphocytes in Treating Patients With Recurrent or Refractory High-Grade Malignant Glioma	Completed	I	Central Nervous System Tumors	3	Therapeutic autologous lymphocytes	NA
68	NCT00331526	Cellular Adoptive Immunotherapy in Treating Patients With Glioblastoma Multiforme	Completed	II	Central Nervous System Tumors	83	Aldesleukin, Therapeutic autologous lymphocytes, Adjuvant therapy, Surgery	USA
69	NCT00004024	Biological Therapy Following Surgery and Radiation Therapy in Treating Patients With Primary or Recurrent Astrocytoma or Oligodendroglioma	Completed	II	Central Nervous System Tumors	60	Aldesleukin, Autologous tumor cell vaccine, Muromonab-CD3, Sargramostim, Therapeutic autologous lymphocytes, Surgical procedure, Radiation therapy	USA

A: Australia; ALECSAT: Autologous Lymphoid Effector Cells Specific Against Tumor; AU: Austria; CHN: China; CTL: cytotoxic T-lymphocytes; DE: Denmark; EGFR BATs: EGFR Bi-armed Activated T-cells; EGFR: epidermal growth factor; EGFRvIII: epidermal growth factor receptor variant III; ES: Spain; FR: France; GBM: Glioblastoma; HER2(EQ)BBζ/CD19t+ Tcm: preparation of genetically modified autologous central memory enriched T-cells (Tcm) expressing a chimeric antigen receptor consisting of an anti-human epidermal growth factor 2 (HER2) variable fragment that is linked to the signaling domain of the T-cell antigen receptor complex zeta chain (BBζ), and truncated cluster of differentiation (CD)19; IL-13Rα2: interleukin-13 receptor α2; IR: Iran; J: Japan; NA: Not Available; PBL: peripheral blood lymphocytes; PD-L1 CSR: programmed death Ligand 1 chimeric switch receptor; RT: Radiotherapy; SE: Sweden; TIL: Tumor-infiltrating T-Lymphocyte; TMZ: temozolomide; USA: United States; ZH: Zurich.

**Table 3 brainsci-11-00386-t003:** Classification of immunotherapies for malignant brain tumors.

Immunotherapies			
Active	Checkpoint Inhibitors	Alkylating agent	TMZ
		MAbs	BVZ, Nivolumab
		Vaccine	Rindopepimut DCVax-Brain PerCellVac2
Adoptive	T cell	TCR transgenic T	
		CAR T	EGFRIII IL-13Ra2 HER2 EphA2
	NK cell	Allogenic NK	
		Anti-KIR Abs	
		ADCC	
		DNRII	
		NK exosomes	
	NKT	Autologous NKT, autologous mature DC cultured with NKT ligand α-galactosyl ceramide	
	Hybrid	ALECSAT	

ADCC: antibody-dependent cellular cytotoxicity; ALECSAT: Autologous Lymphoid Effector Cells Specific Against Tumor; Anti-KIR Abs: Antibody-mediated blocking of KIR; BVZ: Bevacizumab; CAR T: chimeric antigen receptor; DC: Dendritic Cell; DNRII: dominant-negative receptor II; EGFRIII: epidermal growth factor receptor variant III; EphA2: erythropoietin-producing hepatocellular carcinoma A2; HER2: human epidermal growth factor 2; IL-13Ra2: interleukin-13 receptor α2; MAbs: Monoclonal Antibodies; NK: natural killer cells; NKT: T lymphocyte-natural killer cells; T: T lymphocyte; TMZ: Temozolomide.

## Data Availability

All data are included in the main text.

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
