# Peer review of "Against the Resilience of High-Grade Gliomas: The Immunotherapeutic Approach (Part I)"

_brainsci, 2021, doi:10.3390/brainsci11030386_

Round 1
Reviewer 1 Report
Review summary:
In this article titled “Against the Resilience of High-Grade Gliomas: The Immuno-therapeutic Approach (Part I)” authors have reviewed the published literature on this topic and presented a brief introduction on immunotherapeutic agents. The manuscript is well-written and orients the readers on this treatment modality. I support the publication of this manuscript with the following changes:
- Line 35: “Early metastatic spreading”. The term “Metastasis” is typically used for diffuse dissemination in the body from a primary tumor. Extracranial metastasis is not commonly observed in GBM and is a very rare phenomenon which should be mentioned here.
- Line 301-302: “BVZ is indicated in the treatment of recurrent HGGs, but not yet within the upfront 301 protocol for the newly diagnosed recurrent HGGs.” It should be changed to “BVZ is indicated in the treatment of recurrent HGGs, but not yet within the upfront protocol for the newly diagnosed HGGs.”
- Blood brain barrier poses a great impediment to the entry and uptake of immunotherapeutic agents at the site of action and various strategies such as Selective intra-arterial injections after mannitol infusion, focused ultrasonic disruption of BBB prior to delivery, or Surgical innovations such as the use of a temporal fascia flap to bypass the BBB have been utilized. I suggest the authors to add these in the limitations. Here are some related studies on this topic:
- D'Amico RS, et al. Super selective intra-arterial cerebral infusion of modern chemotherapeutics after blood-brain barrier disruption: where are we now, and where we are going [published correction appears in J Neurooncol. 2020 Mar 13;:]. J Neurooncol. 2020;147(2):261–278. doi:10.1007/s11060-020-03435-6
- Etame AB, Diaz RJ, Smith CA, Mainprize TG, Hynynen K, Rutka JT. Focused ultrasound disruption of the blood-brain barrier: a new frontier for therapeutic delivery in molecular neurooncology. Neurosurg Focus. 2012;32(1):E3. doi:10.3171/2011.10.FOCUS11252
- Patel NV, et al. Vascularized Temporoparietal Fascial Flap: A Novel Surgical Technique to Bypass the Blood-Brain Barrier in Glioblastoma [published online ahead of print, 2020 Jul 23]. World Neurosurg. 2020;143:38-45. doi:10.1016/j.wneu.2020.07.132
- D'Amico RS, Neira JA, Yun J, Alexiades NG, Banu M, Englander ZK, Kennedy BC, Ung TH, Rothrock RJ, Romanov A, Guo X, Zhao B, Sonabend AM, Canoll P, Bruce JN. Validation of an effective implantable pump-infusion system for chronic convection-enhanced delivery of intracerebral topotecan in a large animal model. J Neurosurg. 2019 Aug 2:1-10. doi: 10.3171/2019.3.JNS1963. Epub ahead of print. PMID: 31374547; PMCID: PMC7227320.
Author Response
Against the Resilience of High-Grade Gliomas: The Immunotherapeutic Approach (Part I)
Response to Reviewer
We want to thank the kind Reviewer for comments and suggestions that have been precious for us in order to improve the quality and clarity of our manuscript.
We have made substantial revisions of our manuscript and, below, we have reported an itemized, point-by-point response to the Reviewer remarks.
All the changes in the text have been reported in track change mode ON.
Reviewer #1
- “In this article titled “Against the Resilience of High-Grade Gliomas: The Immuno-therapeutic Approach (Part I)” authors have reviewed the published literature on this topic and presented a brief introduction on immunotherapeutic agents. The manuscript is well-written and orients the readers on this treatment modality.”
We want to thank the kind Reviewer for this this comment which made us proud.
- Line 35: “Early metastatic spreading”. The term “Metastasis” is typically used for diffuse dissemination in the body from a primary tumor. Extracranial metastasis is not commonly observed in GBM and is a very rare phenomenon which should be mentioned here.
Thank you for this suggestion, based on which we have modified the sentence in line 35 as follows: “Intrinsic glioma cell heterogenicity, high mitotic activity, abnormal angiogenesis, and early local recurrence are responsible for the resilience of these tumors toward standard treatments”.
Line 301-302: “BVZ is indicated in the treatment of recurrent HGGs, but not yet within the upfront protocol for the newly diagnosed recurrent HGGs.” It should be changed to “BVZ is indicated in the treatment of recurrent HGGs, but not yet within the upfront protocol for the newly diagnosed HGGs.”
We apologize for this typo error. We have changed the sentence as suggested.
“Blood brain barrier poses a great impediment to the entry and uptake of immunotherapeutic agents at the site of action and various strategies such as Selective intra-arterial injections after mannitol infusion, focused ultrasonic disruption of BBB prior to delivery, or Surgical innovations such as the use of a temporal fascia flap to bypass the BBB have been utilized. I suggest the authors to add these in the limitations. Here are some related studies on this topic:
D'Amico RS, et al. Super selective intra-arterial cerebral infusion of modern chemotherapeutics after blood-brain barrier disruption: where are we now, and where we are going [published correction appears in J Neurooncol. 2020 Mar 13;:]. J Neurooncol. 2020;147(2):261–278. doi:10.1007/s11060-020-03435-6
Etame AB, Diaz RJ, Smith CA, Mainprize TG, Hynynen K, Rutka JT. Focused ultrasound disruption of the blood-brain barrier: a new frontier for therapeutic delivery in molecular neurooncology. Neurosurg Focus. 2012;32(1):E3. doi:10.3171/2011.10.FOCUS11252
Patel NV, et al. Vascularized Temporoparietal Fascial Flap: A Novel Surgical Technique to Bypass the Blood-Brain Barrier in Glioblastoma [published online ahead of print, 2020 Jul 23]. World Neurosurg. 2020;143:38-45. doi:10.1016/j.wneu.2020.07.132
D'Amico RS, Neira JA, Yun J, Alexiades NG, Banu M, Englander ZK, Kennedy BC, Ung TH, Rothrock RJ, Romanov A, Guo X, Zhao B, Sonabend AM, Canoll P, Bruce JN. Validation of an effective implantable pump-infusion system for chronic convection-enhanced delivery of intracerebral topotecan in a large animal model. J Neurosurg. 2019 Aug 2:1-10. doi: 10.3171/2019.3.JNS1963. Epub ahead of print. PMID: 31374547; PMCID: PMC7227320.”
We are particularly grateful to the kind Reviewer for this paramount advice, according to which we have added in the Discussion a paragraph titled “Limitations and Future Perspectives” (lines 401-422).
“Several aspects are responsible for the resilience of glioma, such as the lack of tumor antigen, loss of immunological phenotype, and immune evasive molecule production, which all together still limit the success of both active and adoptive immunotherapies.
Furthermore, the main limitation lies in the need to overcome the BBB, reason why the administration route is pivotal. Several studies proposed innovative methods, such as the implanted intracerebral convection-enhanced deliveries or the superselective intra‐arterial cerebral infusion [156-161]. The intra-arterial administration is additionally facilitated by the previous destruction of BBB, carried out through osmotic agents, hypertonic solutions and mannitol, or ultrasounds [156,162,163]. Low frequency ultrasounds are a non-invasive strategy which leads to BBB disruption resulting in increased drug penetration and faster reaching tumor site [162]. An innovative emerging technique exploits the vascularized temporoparietal fascial flaps with the aim to bypass the BBB. These flaps are vascularized by the external carotid artery system, free from the BBB system. They were transposed into the surgical cavity, after glioma resection, and could allow an effective drug penetration and residual tumor cells targeting [164].
Another key aspect is the engineering of small carriers, biocompatible, and non-toxic, able to improve drug diffusion and tissue distribution. Viral vehicles, nano-particles, and liposomes are currently the most investigated [165-168].
The development of new administration routes and the advance in engineering more efficient and safe carriers will allow the implementation of the standard therapeutic protocols with concomitant tailored immunotherapies.”
Furthermore, we have added the suggested articles in the References.
We want to thank once again the kind Reviewer for the valuable suggestions which have been paramount for us to improve the overall clarity and quality of the manuscript.
The Authors
Reviewer 2 Report
I think that is an ambitious project to try to summarize some of the immunotherapeutic studies done in glioblastoma research. Often, reports focus on a selected part of the field instead.
Materials and Methods
The review is, as any review, limited to cover papers by the chosen key words. Some key words are quite specific. Why? It is also stated that the wider search with only the word immunotherapy was used. It is not quite clear why the authors first have very specific key words.
+ the review was performed in a systematic way using RPISMA protocol.
Results
” implementation of the exclusion criteria” - which are the exclusion criteria?
” Table 1 summarizes the main clinical trials on immunotherapies for 76 HGGs (Table 1)” – does this mean that only clinical trials are considered? This needs to be stated much more clear in the materials and methods section…
What Table 1 actually presents is the small variation of drugs that are tested in the majority of the clinical trials. In this way, I think that the table conveys an important message.
BBB passage is described for most compounds, but not monoclonal antibodies; should be mentioned what is known/not known.
What are the results of the studies presented? I think that this information should be included as well. How was overall survival affected by the treatment?
Discussion
Should cover – does standard therapy reduced the immune response? Problems trying to deliver immune therapy in conjunction with standard therapy.
Author Response
Against the Resilience of High-Grade Gliomas: The Immunotherapeutic Approach (Part I)
Response to Reviewer
We want to thank the kind Reviewer for comments and suggestions that have been precious for us in order to improve the quality and clarity of our manuscript.
We have made substantial revisions of our manuscript and, below, we have reported an itemized, point-by-point response to the Reviewer remarks.
All the changes in the text have been reported in track change mode ON.
Reviewer #2
- “I think that is an ambitious project to try to summarize some of the immunotherapeutic studies done in glioblastoma research. Often, reports focus on a selected part of the field instead.”
We agree with the Reviewer with the difficulty found in the attempt to summarize the immunotherapeutic studies on glioblastoma. Indeed, the aim of our review was to analyze and present the results of the main studies, especially those which have been seminal papers in the field. This with the idea to allow the reader to have a synopsis that may be a practical tool and a guide for those who preliminary face with this fascinating but complex approach to malignant gliomas, in line with the main aim of the special issue.
- “Materials and Methods
The review is, as any review, limited to cover papers by the chosen key words. Some key words are quite specific. Why? It is also stated that the wider search with only the word immunotherapy was used. It is not quite clear why the authors first have very specific key words.
+ the review was performed in a systematic way using PRISMA protocol.”
Thank you very much for this point which, rightly, is worthy of a clearer explanation. The literature search has been conducted according to PRISMA guidelines and the searching process has been more articulated than reported in the previous version. We have started with more generic terms as “immunotherapy, active”; “immunotherapy, adoptive”, to progressively restrict the field to more specific keywords as “vaccine”; “alkylating agents”; “monoclonal antibodies”; “engineered T cell”; and “allogenic NK cell”. The choice of these highly specific keywords has derived from the need to focus on equally specific aspects of the immunotherapies, as rightly highlighted by the kind Reviewer. A further reason at the base of the use of the aforementioned keywords lies in the fact that the same fixed terms have been selected within the Mesh database of Pubmed, which played as a valuable tool for this literature review.
According to the suggestions of the Reviewer, we have clarified these aspects in Material and Methods as follows: “(lines 62-74): “A comprehensive online literature review was performed in line with the Preferred Reporting Items for Systematic Reviews and Meta-Analysis (PRISMA) guidelines. The PubMed/Medline (https://pubmed.ncbi.nlm.nih.gov) and ClinicalTrials.gov (https://clinicaltrials.gov) databases were used, with combinations of Medical Subject Headings (MeSH) terms and text words. The main MeSH terms and key words were “malignant brain tumor,” “high-grade glioma,” and “glioblastoma”, furtherly merged with “immunotherapy, active”; “immunotherapy, adoptive”. Supplementary research was conducted with additional MeSH terms: “chemotherapy”; “vaccine”; “alkylating agents”; “monoclonal antibodies”; “engineered T cell”; and “allogenic NK cell”, in order to restrict the field of interest to the novel immunotherapeutic strategies. The eligibility criteria included only articles written in English or translated, published in the last 10 years, and related to neuro-oncology. Review articles and editorials were included and filtered according to best match and relevance based on title and abstract.”.
Furthermore, in order the better describe the selection process of the articles, we have clarified the Inclusion and Exclusion Criteria in the Table 1 of the new edited versions of the manuscript. (lines 84-85):
Table 1. Inclusion and Exclusion Criteria for Systematic Review
|
Inclusion Criteria |
Exclusion Criteria |
|
Reviews, Peer-Reviews, Editorials |
Case Reports, Abstracts, and Dissertations |
|
Clinical, Pre-clinical Trials |
Withdrawn or Abandoned Clinical Trials |
|
English language, or translated |
Non-English language |
|
Publications in 2010-2020 decade |
Studies out of 2010-2020 decade |
|
Studies on Human, or Human Products |
Animal Studies |
|
Publications related to neuro-oncology |
Publications not related to neuro-oncology |
|
Publications related to High-Grade Glioma |
Publications not related to High-Grade Glioma |
|
Adult and pediatric patients |
Accordingly, we have updated and modified the Figure 1 (lines 93-94).
Results
“implementation of the exclusion criteria” - which are the exclusion criteria?”
On the basis of this suggestion, we have added in Materials and Methods Table 1, which summarizes all the inclusion and exclusion criteria for PRISMA-based systematic review (lines 84-85): “All the inclusion and exclusion criteria are outlined in Table 1 (Table 1).”
“Table 1 summarizes the main clinical trials on immunotherapies for HGGs (Table 1)” – does this mean that only clinical trials are considered? This needs to be stated much more clear in the materials and methods section…
Thank you for this point. Yes, in Table 1 we have focused only on the Interventional studies and clinical trials extracted from the ClinicalTrials.gov database. The reason for having dedicated Table 1 (Table 2 in the new version) specifically to clinical trials lies in the wide relevance of this type of studies.
We have better clarified these aspects in the Materials and Methods (lines 75-82): “On the ClinicalTrials.gov database, the search terms used to identify clinical trials were as follows: “malignant brain tumor,” “high-grade glioma” “central nervous system”, “immunotherapy, active”, and “immunotherapy, adoptive”. Interventional studies and clinical trials were included. No limits for the study phase or recruitment status were applied. Duplicates and titles with no English language translation available were removed. Trials related to innovative therapies for high-grade gliomas were chosen.”
What Table 1 actually presents is the small variation of drugs that are tested in the majority of the clinical trials. In this way, I think that the table conveys an important message.
We agree with the important message of the table on clinical trials. Actually, through the rightly highlighted small variation existing among the drugs tested, it is possible to draw a tendency toward specific lines of evidence-based approaches within the complex panorama of immunotherapies, where TMZ and monoclonal antibodies have been the more tested.
- BBB passage is described for most compounds, but not monoclonal antibodies; should be mentioned what is known/not known.
Thank you very much also for this important suggestion.
We agree with the need to integrate the results with the passage of the monoclonal antibodies though the blood-brain-barrier.
We added in the Results, paragraph “Monoclonal Antibodies”, the following part: (lines 238-245): “Despite BVZ and nivolumab were studied for glioma therapy in several clinical trials and also as a combined protocol (#NCT03890952, #NCT03743662, #NCT03452579) [68-70], the administration route remains still a concern. Because of the BBB physiologically blocks the antibody access to the brain, several studies are focusing on innovative mAbs delivering routes, to improve the drug efficacy and intratumoral uptake. Intraarterial administration, intracranial injection, nanoparticle and liposomal carriers are currently the more potential strategies [56,71-74].”
In addition, we have added in the Discussion the paragraph “Limitations and Future Perspectives” (lines 401-422), where we have also deepened the mechanisms of blood-brain barrier passage and drug carriers: “Several aspects are responsible for the resilience of glioma, such as the lack of tumor antigen, loss of immunological phenotype, and immune evasive molecule production, which all together still limit the success of both active and adoptive immunotherapies.
Furthermore, the main limitation lies in the need to overcome the BBB, reason why the administration route is pivotal. Several studies proposed innovative methods, such as the implanted intracerebral convection-enhanced deliveries or the superselective intra‐arterial cerebral infusion [156-161]. The intra-arterial administration is additionally facilitated by the previous destruction of BBB, carried out through osmotic agents, hypertonic solutions and mannitol, or ultrasounds [156,162,163]. Low frequency ultrasounds are a non-invasive strategy which leads to BBB disruption resulting in increased drug penetration and faster reaching tumor site [162]. An innovative emerging technique exploits the vascularized temporoparietal fascial flaps with the aim to bypass the BBB. These flaps are vascularized by the external carotid artery system, free from the BBB system. They were transposed into the surgical cavity, after glioma resection, and could allow an effective drug penetration and residual tumor cells targeting [164].
Another key aspect is the engineering of small carriers, biocompatible, and non-toxic, able to improve drug diffusion and tissue distribution. Viral vehicles, nano-particles, and liposomes are currently the most investigated [165-168].
The development of new administration routes and the advance in engineering more efficient and safe carriers will allow the implementation of the standard therapeutic protocols with concomitant tailored immunotherapies.”
What are the results of the studies presented? I think that this information should be included as well. How was overall survival affected by the treatment?
We agree also with the necessity to better report how the overall survival and progression-free survival of GBM has been affected by these therapies.
As a consequence, we have highlighted the results of the main studies on mortality as follows:
- Lines 228-233: “The AVAglio92 documented a significant improvement of the median progression free survival (PFS) to 10.6 months in the BVZ group, versus 6.2 months in the placebo one (p < 0.001). Overall survival (OS) proved to be higher only during the first two years of treatment in the BVZ group (72.4% versus 66.3%) [63]. The RTOG-082593 reported an increase PFS in the BVZ group than the control group (10.7 months versus 7.3 months, p=0.007), but equally failed to demonstrate a better OS [64].”.
- Lines 255-256: “The first phase I trial, VICTORI, showed an excellent safety profile, a PFS of 10.2 months and an OS of 22.8 months.”
- Lines 258-265: “The ACTIVATE phase II trial reported a PFS of 14.2 and 6.4 months, and an OS of 26 and 15.2 months in the vaccinated patients and control group, respectively. In ACT II and III studies, the PFS was 15.2 and 12.3 for the rindopepimut group, respectively, while the PFS was 6.4 months in both control groups. The OS was 23.6 and 24.6 months in ACT II and III trials, respectively; compared to the control groups (15.2 months). ReACT showed a PFS at-6months of 28% for rindopepimut, compared to 16% in the control group [79,81-84].”
- Lines 266-268: “ACT IV, a multicentric phase III trial, tested the combination of rindopepimut and standard chemotherapy with TMZ in newly diagnosed GBM, showing no significant difference in OS between the two groups (#NCT01480479) [85].”
- Line 299-303: “In 2017, O’Rourke et colleagues treated 10 patients with a single-dose of EGFRvIII CAR T cells (#NCT02209376) reporting no dose-related side effects and a good safety profile, but neither the OS nor the PFS has been shown to increase. In 2017, Ahmed et al. tested HER2-specific CAR T cell for treatment of 17 recurrent HER2+ high-grade gliomas (#NCT01109095) showed similarly results.”
We have also added in the Conclusions the following sentence (line 493-494): “Although they are still not considered to be the first-line treatment against malignant gliomas, immunotherapies have shown excellent results in improve the PFS”.
Discussion
Should cover – does standard therapy reduced the immune response?
Thank you for this interesting question. While admitting the lack of evidence of direct interaction between the two therapies, TMZ is known to potentially cause iatrogenic lymphopenia, which would decrease furtherly the immune response against the tumor, at least theoretically.
We have clarified this aspect adding in the Discussion the following sentence: (lines 363-366) “TMZ also frequently induces lymphopenia and myelosuppression, as many other chemotherapy drugs. The immunodepletion leads to less immune surveillance and inhibition of antitumor immune response [136-138].”.
- Problems trying to deliver immune therapy in conjunction with standard therapy.
Thank you also for this point. Indeed, the delivery of these drugs is one of the main problems to urgently overcome of these therapies, to the point that we have realized the need to dedicate a large part of the paragraph titled “Limitations and Future Perspectives”, specifically to this aspect: (lines 401-422) “Several aspects are responsible for the resilience of glioma, such as the lack of tumor antigen, loss of immunological phenotype, and immune evasive molecule production, which all together still limit the success of both active and adoptive immunotherapies.
Furthermore, the main limitation lies in the need to overcome the BBB, reason why the administration route is pivotal. Several studies proposed innovative methods, such as the implanted intracerebral convection-enhanced deliveries or the superselective intra‐arterial cerebral infusion [156-161]. The intra-arterial administration is additionally facilitated by the previous destruction of BBB, carried out through osmotic agents, hypertonic solutions and mannitol, or ultrasounds [156,162,163]. Low frequency ultrasounds are a non-invasive strategy which leads to BBB disruption resulting in increased drug penetration and faster reaching tumor site [162]. An innovative emerging technique exploits the vascularized temporoparietal fascial flaps with the aim to bypass the BBB. These flaps are vascularized by the external carotid artery system, free from the BBB system. They were transposed into the surgical cavity, after glioma resection, and could allow an effective drug penetration and residual tumor cells targeting [164].
Another key aspect is the engineering of small carriers, biocompatible, and non-toxic, able to improve drug diffusion and tissue distribution. Viral vehicles, nano-particles, and liposomes are currently the most investigated [165-168].
The development of new administration routes and the advance in engineering more efficient and safe carriers will allow the implementation of the standard therapeutic protocols with concomitant tailored immunotherapies.”
We want to thank once again the kind Reviewer for the valuable suggestions which have been paramount for us to improve the overall clarity and quality of the manuscript.
We hope that this new edited version of our manuscript may have improved its quality.
The Authors.